# Antimicrobial Chlorinated 3-Phenylpropanoic Acid Derivatives from the Red Sea Marine Actinomycete *Streptomyces*
*coelicolor* LY001

**DOI:** 10.3390/md18090450

**Published:** 2020-08-27

**Authors:** Lamiaa A. Shaala, Diaa T. A. Youssef, Torki A. Alzughaibi, Sameh S. Elhady

**Affiliations:** 1Natural Products Unit, King Fahd Medical Research Center, King Abdulaziz University, Jeddah 21589, Saudi Arabia; 2Department of Medical Laboratory Sciences, Faculty of Applied Medical Sciences, King Abdulaziz University, Jeddah 21589, Saudi Arabia; taalzughaibi@kau.edu.sa; 3Suez Canal University Hospital, Suez Canal University, Ismailia 41522, Egypt; 4Department of Natural Products, Faculty of Pharmacy, King Abdulaziz University, Jeddah 21589, Saudi Arabia; ssahmed@kau.edu.sa; 5Department of Pharmacognosy, Faculty of Pharmacy, Suez Canal University, Ismailia 41522, Egypt; 6King Fahd Medical Research Center, King Abdulaziz University, Jeddah 21589, Saudi Arabia

**Keywords:** Red Sea sponges, marine actinomycetes, *Streptomyces**coelicolor* LY001, halogenated 3-phenylpropanoic acid derivatives, diketopiperazine alkaloids, structural determinations, antimicrobial activities

## Abstract

The actinomycete strain *Streptomyces coelicolor* LY001 was purified from the sponge *Callyspongia siphonella*. Fractionation of the antimicrobial extract of the culture of the actinomycete afforded three new natural chlorinated derivatives of 3-phenylpropanoic acid, 3-(3,5-dichloro-4-hydroxyphenyl)propanoic acid (**1**), 3-(3,5-dichloro-4-hydroxyphenyl)propanoic acid methyl ester (**2**), and 3-(3-chloro-4-hydroxyphenyl)propanoic acid (**3**), together with 3-phenylpropanoic acid (**4**), *E*-cinnamic acid (**5**), and the diketopiperazine alkaloids cyclo(l-Phe-*trans*-4-OH-l-Pro) (**6**) and cyclo(l-Phe-*cis*-4-OH-d-Pro) (**7**) were isolated. Interpretation of nuclear magnetic resonance (NMR) and high-resolution electrospray ionization mass spectrometry (HRESIMS) data of **1**–**7** supported their assignments. Compounds **1**–**3** are first candidates of the natural chlorinated phenylpropanoic acid derivatives. The production of the chlorinated derivatives of 3-phenylpropionic acid (**1**–**3**) by *S. coelicolor* provides insight into the biosynthetic capabilities of the marine-derived actinomycetes. Compounds **1**–**3** demonstrated significant and selective activities towards *Escherichia. coli* and *Staphylococcus aureus*, while *Candida albicans* displayed more sensitivity towards compounds **6** and **7**, suggesting a selectivity effect of these compounds against *C. albicans*.

## 1. Introduction

The marine actinomycetes represent a vital source of biologically active secondary metabolites and a promising future source for drug discovery. It is well known that marine tunicates and sponges are highly associated with symbiotic microbes [1,2,3]. There are very few reports about investigation of Red Sea actinomycetes for their chemical diversity and biomedical importance [4].

Streptomycetes represent a group within the actinomycetes with an economical importance and represent a vigorous source of different bioactive secondary metabolites [5]. More than 75% of the marketed antibiotics, commercially available compounds and several agrochemicals are synthesized by streptomycetes [6,7,8]. In 1990, the genus *Streptomyces* alone represented the main source of almost 60% of the antibiotics and most of the agrochemicals [9]. Streptomycetes produce an array of diverse and bioactive compounds with antimicrobial [5,9,10,11], anticancer [5,12,13], insecticidal and antiparasitic [14], anti-inflammatory [15], anti-fouling [16], antiviral and anti-infective [17,18] properties. Moreover, the genus *Streptomyces* is considered as a candidate of industrial importance [19,20,21], and a producer of secondary metabolites with herbicidal activity and which promote plant growth [22], vitamins [23], ribonucleases [11,13,24,25,26,27], and enzyme inhibitors [13]. These features make these microorganisms an ideal and favorite research project for academia and industry [9]. Recent advances in marine microbiology, including their purification and identification of bacteria that produce rich arrays of bioactive natural products, provide a strong motivation to explore aggressively their potential as sources of novel pharmaceutical agents [28,29,30]. Likewise, new methodology for rapid and affordable sequencing of microbial chromosomes, bioinformatic and metabolic profiling enables information on the predicted secondary metabolic diversity contained within a bacterial genome to be rapidly evaluated and manipulated [31,32,33].

Due to the fast growing number of pathogenic microbes, viral infections and cancer cells that have resistance towards current therapies, drug leads, and new chemical entities for the enhancement of new drugs are in high demand. Tuberculosis, malaria, and *Staphylococcus aureus* infections are among just a few diseases that have become difficult to treat with antibiotics [34]. There is, therefore, a pressing need for the development of new methodologies to provide new drugs/drug leads for the future. Marine microorganisms, such as actinomycetes that exist in association with marine invertebrates and algae, produce novel chemical entities with potential pharmaceutical significance. The importance of natural products, however, extends far beyond that of drug discovery to that of addressing fundamental biological questions, taking full advantage of their structural complexity and functional diversity to probe biological function.

The decreasing number of FDA approved therapeutics, and the low number of drugs in the pipeline that have arisen from synthetic combinatorial libraries, has spurred renewed interest in natural products research with the aim of identifying new structural scaffolds. Marine-derived actinomycetes represent a promising and vigorous source of ubiquitous and diverse and bioactive secondary metabolites that can be obtained in large-scale cultures and developed as drug leads. Overall, there are relatively few reports of marine microbial cultivation from the highly biodiverse Red Sea. The results from this study confirm that Red Sea marine actinomycetes represent a promising source of new drug leads with antibiotic potential.

As a part of our endeavor to purify and characterize bioactive marine microbial candidates [35,36], the antimicrobial fractions of extract of the culture of *Streptomyces coelicolor* LY001 was investigated. Three chlorinated derivatives of 3-phenylpropanoic acid, including 3-(3,5-dichloro-4-hydroxyphenyl)propanoic acid (**1**), 3-(3,5-dichloro-4-hydroxyphenyl)propanoic acid methyl ester (**2**), and 3-(3-chloro-4-hydroxyphenyl)propanoic acid (**3**), along with 3-phenylpropanoic acid (**4**) [37], *E*-cinnamic acid (**5**) and the diketopiperazine alkaloids cyclo(l-Phe-*trans*-4-OH-l-Pro) (**6**) [38,39], and cyclo(l-Phe-*cis*-4-OH-d-Pro) (**7**) [40,41] were isolated and identified. Compounds **1**–**7** were determined by assignments of their NMR and HRESIMS data. Herein, the isolation, structure assignments, and the antimicrobial activities of **1**–**7** are presented.

## 2. Results and Discussion

### 2.1. Isolation of the Actinomycete, Streptomyces coelicolor LY001

The marine-derived actinomycete LY001 (Figure 1) was isolated from the internal tissues of the sponge *Callyspongia siphonella* (Figure 1). The resulting sequence of the actinomycete strain was searched for homology with Basic Local Alignment Search Tool (BLAST) in the GenBank. The alignment with reported sequences in the GenBank showed that the LY001 isolate belongs to the genus *Streptomyces* and displayed 100% similarity with the strain *Streptomyces coelicolor* AB588124.

### 2.2. Structure Elucidation of the Compounds

The structural assignment of compound **1** (Figure 2) was supported by interpretation of its NMR spectra (Appendix A). The molecular formula of C_9_H_8_Cl_2_O_3_ was suggested for **1** as supported by HRESIMS (234.9932, C_9_H_9_Cl_2_O_3,_ [M + H]^+^) (Appendix A), suggesting five degrees of unsaturation. Its ^13^C NMR spectrum along with the heteronuclear single-quantum correlation spectroscopy (HSQC) experiment showed seven signals equivalent for nine carbons including the equivalent methines (C-2 and C-6), two methylenes (C-7 and C-8) and five quaternary carbons, including two chemically equivalent carbons (C-3 and C-5) (Table 1). Its ^1^H NMR demonstrated a 2-proton singlet at δ_H_ 7.12 for the chemically equivalent protons H-2 and H-6 (Table 1). This two-proton singlet at δ_H_ 7.12 was correlated to the signal at δ_C_ 128.1 (CH, C-2/C-6) in the HSQC experiment. In the ^1^H-^1^H correlation spectroscopy (COSY) spectrum, vicinal couplings between the methylene protons at δ_H_ 2.86 (2H, H_2_-7) and 2.64 (2H, H_2_-8) was observed. Further, the protons of the methylenes at δ_H_ 2.86 (H_2_-7) and 2.64 (H_2_-8) were correlated, in the HSQC experiment, to the ^13^C NMR signals at δ_C_ 39.3 (CH_2_, C-7) and 34.5 (CH_2_, C-8), respectively. In addition, the existence of the quaternary carbons at δ_C_ 133.6 (qC, C-1), 122.1 (qC, C-3, C-5), and 146.0 (qC, C-4) together with δ_C_ 128.1 (2 × CH, C-2/C-6) suggested a 1,3,4,5-tetrasubstituted benzene moiety with OH at C-4 and symmetrical substitutions with chlorine atoms at C-3 and C-5.

^1^H-^13^C Heteronuclear multiple bond correlation spectroscopy (HMBC) experiment supported the substitution on the phenyl moiety. HMBC of H-2,6/C-1 (qC, δ_C_ 133.6), H-2,6/C-3,5 (qC, δ_C_ 122.1), and H-2,6/C-4 (qC, δ_C_ 146.9) supported this substitution (Figure 3). Thus, 3,5-dichloro-4-hydroxyphenyl moiety was assigned as part A of the molecule (Figure 2). Furthermore, the ^1^H chemical shifts values of H-2 and H-6, and ^13^C values of C-1–C-6 in **1** are similar with those of (*S*)-2-amino-3-(3,5-dichloro-4-hydroxyphenyl)-propanoic acid [42]. The substituent at C-1 of the phenyl moiety was assigned as 3-substituted propanoic acid (part B) as supported from the signals at δ_H/C_ 2.86 (2H)/29.3 (H_2_-7/C-7), 2.64 (2H)/34.5 (H_2_-8/C-8), and δ_C_ 173.8 (qC, C-9) (Table 1). The HMBC from the protons at H_2_-7 and H_2_-8 to C-9 (δ_C_ 173.8) (Figure 3) supported this assignment. Finally, the connection between the two parts (A and B) of **1** was proved by cross-peaks in the HMBC of H_2_-7/C-1, H_2_-7/C-2,6, H_2_-8/C-1, and H-2,6/C-7 (Figure 3). Thus, compound **1** was assigned as 3-(3,5-dichloro-4-hydroxyphenyl)propanoic acid and this is the first report about its natural occurrence.

Interpretation of the one- and two-dimensional NMR data (Appendix A) supported the structural determination of **2** (Figure 2). It possesses molecular formula C_10_H_10_Cl_2_O_3_ as supported by HRESIMS (249.0088, C_10_H_11_Cl_2_O_3_, [M + H]^+^) (Appendix A), being 15 mass units larger than **1**, proving the existence of an additional CH_3_ in **2**. The structure of **2** was assigned by interpretation of its NMR spectra. The NMR data of **2** are identical with those of **1** (Table 1), suggesting similar structures. Furthermore, the existence of an additional three-proton singlet at δ_H_ 3.67 correlated to the signal at δ_C_ 51.9 (CH_3_, C-10) in the HSQC, which supports the existence of a terminal methoxyl group in **2**. The HMBC from H_3_-10 (δ_H_ 3.67) to C-9 (δ_C_ 173.3) supported the existence of a methyl ester group in **2** instead of a free carboxylic acid in **1**.

Further, the assignment of the substituents on the aromatic moiety was supported by HMBC (Figure 3) as previously discussed under compound **1**. Similarly, the ^1^H chemical shifts of H-2 and H-6, and ^13^C chemical shifts values of C-1–C-6 in **2** are similar to those of (*S*)-2-amino-3-(3,5-dichloro-4-hydroxyphenyl)-propanoic acid [42]. Accordingly, compound **2** was assigned as 3-(3,5-dichloro-4-hydroxyphenyl)propanoic acid methyl ester and this is its first natural occurrence.

The structure determination of compound **3** (Figure 2) was assigned by examination of its NMR spectroscopic data (Appendix A). Compound **3** with molecular formula of C_9_H_9_ClO_3_ (201.0321, C_9_H_10_ClO_3_, [M + H]^+^) (Appendix A), being 34 units less than **1**, suggests the presence of only one chlorine atom in **3**. Its ^13^C NMR spectrum and HSQC exhibited nine signals for three aromatic methines (C-2, C-5, and C-6), two methylenes (C-7 and C-8), and four quaternary carbons (Table 1). Its ^1^H NMR spectrum showed two spin-spin coupling systems. The first one includes the aromatic ABX coupling system at δ_H_ 7.17 (H-2, d, *J* = 2.5 Hz), 6.93 (H-5, d, *J* = 8.5 Hz), and 7.02 (H-6, dd, *J* = 8.5, 2.5 Hz) (Table 1). These protons are correlated in the HSQC experiment to the signals at δ_C_ 128.6 (C-2), 116.1 (C-5), and 128.4 (C-6), respectively. This system supported the existence of a 1,3,4-trisubstituted aromatic moiety in **3**. The second spin coupling system includes the vicinal coupling between the methylenes at δ_H_ 2.86 (H_2_-7, t, *J* = 7.6 Hz) and 2.65 (H_2_-8, t, *J* = 7.6 Hz). In the HMBC, H_2_-7 (δ_H_ 2.86) and H_2_-8 ((δ_H_ 2.65) displayed correlation to the signal at δ_C_ 173.9 (C-9) (Figure 3), supporting the existence of a 3-substituted propanoic acid moiety as in **1**. Again, the placement of the side chain (3-substituted propanoic acid) at C-1 was secured from HMBC of H-2/C-7, H-6/C-7, H_2_-7/C-1, and H_2_-8/C-1 (Figure 3). Finally, HMBC of H-2/C-3 (qC, δ_C_ 120.8), H-2/C-4 (qC, δ_C_ 151.7), H-5/C-4, H-5/C-3, H-5/C-1 (qC, δ_C_ 133.5), H_2_-7/C-9 (qC, δ_C_ 173.9), and H_2_-8/C-9 (Figure 3) supported the substitution on the benzene ring in **3**. Again, the chemical shifts of the ^1^H (H-2, H-5 and H-6) and ^13^C (C-1–C-6) NMR signals of the 3,5-dichloro-4-hydroxy phenyl moiety in compound **3** are similar with reported data for (*S*)-3-(3-chloro-4-hydroxyphenyl)-2-(dimethylamino)propanoic acid [42]. Thus, **3** was assigned as 3-(3-chloro-4-hydroxyphenyl)propanoic acid and this is the first report about its natural occurrence.

Interpretation of the NMR (Appendix A) and MS spectra of **4** (Figure 2) supported its structure. The NMR are identical with those reported for 3-phenylpropanoic acid from a terrestrial *Streptomyces* strain [37]. Accordingly, **4** was assigned as 3-phenylpropanoic acid and this is its first occurrence from a marine *Streptomyces*.

The structure of **5** (Figure 2) was assigned, from the NMR (Appendix A) and MS spectra, as *E*-cinnamic acid. Its NMR spectra exhibited signals for a monosubstituted benzene ring. The *J* value (*J*_7,8_ = 16.1 Hz) supported *E* configuration in **5**.

The structures of **6** and **7** (Figure 2) were identified from interpretation of the corresponding NMR (Appendix A for **6** and Appendix A for **7**) and MS spectra. Both compounds (Figure 2) displayed the same molecular formula of (C_14_H_16_N_2_O_3_). The diketopiperazine nature of **6** and **7** are obvious from the corresponding ^1^H and ^13^C resonating signals (Table 2). Those include the resonances of two amidic carbonyls (C-2 and C-5) as well as the resonances for the methine signals at C-3 and C-6 (Table 2). The remaining of the signals are characteristic for the presence of a hydroxylated proline and phenylalanine moieties in **6** and **7** [38,39,40,41]. The NMR signals at δ_H/C_ 4.60/68.4 (in **6**) and δ_H/C_ 4.40/68.1 (in **7**) are characteristic for the hydroxylated C-8 supporting the hydroxylation of the proline moieties in both compounds [38,39,40,41]. The NMR data of **6** and **7** (Table 2) are similar to those of cyclo(l-Phe-*trans*-4-OH-l-Pro) [38,39] and cyclo(l-Phe-*cis*-4-OH-d-Pro) [40,41], respectively. Furthermore, the 2D experiments (HSQC, COSY, HMBC) (Figure 3) supported the assignment of all signals of the compounds (Table 2 and Figure 3). Accordingly, compounds **6** and **7** were assigned as cyclo(l-Phe-*trans*-4-OH-l-Pro) and cyclo(l-Phe-*cis*-4-OH-d-Pro).

The derivatives of 3-phenylpropanoic acid are rarely reported from microbial organisms. We believe that there is only one report about the occurrence of 3-phenylpropanoic acid from a terrestrial *Streptomyces* strain [37]. Further, it is worth pointing out that this is the first report about the natural existence of chlorinated derivatives of 3-phenylpropanoic acid. These results give an understanding about the biosynthetic potential and structural diversity of the cultured marine-derived microbes and the future application of these metabolites in drug discovery.

Phenylpropanoids area ubiquitous group of organic compounds including *flavonoids*, coumarins, phenolic acids, stilbenes, and lignins. They originate from phenylalanine and tyrosine [43]. The name phenylpropanoid is derived from a six-carbon forming a phenyl moiety connected to a three-carbon propene unit. The coumaric acid represents the key biosynthetic intermediate in the biosynthesis of all phenylpropanoids. Biosynthetically, a variety of natural products including flavonoids, phenylpropanoids, isoflavonoids, catechins, coumarins, stilbenes, aurones, and lignols originates from 4-coumaroyl-CoA [44], which is originated from cinnamic acid [44].

Three pathogens were used to determine the antimicrobial effects and the minimum inhibitory concentration (MIC) values of **1**–**7**. Compound **1** showed the greatest inhibitory effects towards *S. aureus* and *E. coli* with inhibition zones of 17 and 23 mm (Table 3). In addition, compounds **2** and **3** were less active than **1** against these pathogens with inhibition zones of 12–20 mm (Table 3). Compound **4** displayed the lowest activity against these pathogens with inhibition zones of 15 and 11 mm. These findings suggest the importance of the substitution with a “3,5-dichloro-4-hydroxy” moiety on the phenyl moiety as well as the presence of a free terminal carboxylic acid moiety for a maximum antibacterial activity (as in **1**). On the other hand, **1**–**4** showed modest effect towards *C. albicans* (ATCC 14053) with 6–9 mm inhibition zone (Table 3), suggesting selective antibacterial effects of these compounds against *E. coli* and *S. aureus*. Finally, the diketopiperazine alkaloids **6** and **7** displayed better activities towards *C. albicans* with 12–14 mm inhibition zones, while they were less active towards *E. coli* and *S. aureus* with 7–11 mm inhibition zones, suggesting their selective antifungal activity against *C. albicans*.

To determine the MIC values of the compounds, a microdilution method was carried out (Table 3). Compound **1** displayed the highest activity towards *E. coli* with an MIC of 16 µg/mL, while compounds **6** and **7** displayed the highest antifungal activities with an MIC of 32 µg/mL towards *C. albicans*. On the other hand, compounds **2** and **3** displayed lower activities with MIC values of 32–64 µg/mL towards *E. coli* and *S. aureus*. Other compounds were weakly active with MIC values of 64–250 μg/mL.

## 3. Materials and Methods

### 3.1. General Experimental Procedures

Optical rotations, ultraviolet (UV), infra-red (IR), NMR, and HRESIMS were acquired as previously reported [35,36]. Fractionation of the extracts and successive fractions were performed on SiO_2_ and Sephadex LH-20. The purification of the compounds was carried out on an analytical Shim-Pack C18 (250 × 4.6 mm, Shimadzu, Kyoto, Japan).

### 3.2. The Host Organism, C. siphonella

The marine sponge *C. siphonella* was harvested in May 2016 using scuba at a depth up to 20 m off Jizan at the Saudi Red Sea. The pink tubular sponge is dichotomously divided with a smooth thin-walled surface. It possesses a soft compressible consistency, which is difficult to tear. The voucher specimen measures up to 10 cm, while the branching tubes measure up to 5 cm in height and up to 2.5 cm in diameter. A comprehensive description of the sponge and the specimen’s codes are previously reported [45,46].

### 3.3. Isolation of the Actinomycete Streptomyces coelicolor LY001

After surface sterilization, about 1 cm^3^ of the internal tissue of the sponge was finely mixed in sterile seawater (10 mL), diluted, and spread on International Streptomyces Project-2 (ISP2) medium. The medium was amended with 3% NaCl. Afterwards, cultured plates were incubated at 30 °C and checked after actinomycetes growth regularly. The actinomycete LY001 was obtained in a pure state after several purification steps.

### 3.4. Characterization of the Actinomycete, Streptomyces coelicolor

The LY001 strain was identified by analysis of its 16S rRNA sequence. DNA preparation was used for 16S rRNA gene PCR amplification using 27f and 1492r primers. The reaction mixture composed of 50 µL included 1000 ng of gDNA, primers (each 20 pmol), and GoTaq Master Mixes (25 µL). The polymerase chain reaction (PCR) thermocycler initiated with denaturation at 95 °C (2 min), 30 cycles at 95 °C (30 s for denaturation), 30 s at 58 °C (for annealing), 60 s at 72 °C (for extension), and at 72 °C for 5 min for final completion of DNA extension. Purification of PCR products was accomplished on Agarose Gel DNA Purification Kit (Biocompare, South San Francisco, CA, USA) as supported by the supplier. 16S rRNA sequence displayed 100% similarity with *Streptomyces coelicolor* (Accession No. AB588124). The sequence of the Red Sea *Streptomyces coelicolor* LY001 was placed in the NCBI GenBank under the Accession Number MN883509 on 30 December 2019 (http://getentry.ddbj.nig.ac.jp/).

### 3.5. Large-Scale Culture of Streptomyces coelicolor

Spores of *Streptomyces coelicolor* were cultured in 2.0 L flasks, each containing ISP2 media (500 mL) [47] including 10 g of malt extract, 4.0 g of yeast extract, 4.0 g of dextrose, and 3% NaCl (*w*/*v*) in 1 L distilled water at pH of 7. Incubation of the culture was accomplished by shaking at 180 rpm at 28 °C for 14 days. The combined culture broth (10 L) was shaken against EtOAC three times (each 3 L). The resulted EtOAc extracts dried to give 1.3 g.

### 3.6. Purification of ***1***–***7***

The dried extract (1.3 g) was partitioned on C18 (ODS) silica (Sigma Aldrich, St. Louis, MO, USA) using vacuum liquid chromatography (VLC)column using H_2_O-MeOH gradients to give 10 fractions (Fractions A–J). The antimicrobial fractions C (180 mg) and D (118 mg) were separately fractionated on Sephadex LH-20 using MeOH-CH_2_Cl_2_ (1:1) mixture to give five subfractions, each. Subfraction C4 (65 mg) was purified on an ODS HPLC column using 30% CH_3_CN to afford compounds **5** (9.1 mg), **6** (3.7 mg), and **7** (2.6 mg). Subfraction D4 (54 mg) was purified on an ODS HPLC using 45% CH_3_CN to afford compounds **1** (6.2 mg), **2** (3.6 mg), **3** (3.1 mg), and **4** (4.2 mg).

#### Spectral Data of **1**–**7**

3-(3,5-Dichloro-4-hydroxyphenyl)propanoic acid (**1**). Colorless solid; UV (MeOH) λ_max_ (log ε): 227 (4.15), 312 (4.17) nm; IR (film) ν_max_ 3340, 3029, 1700, 1301, 1219, 935 cm^−1^; HRESIMS *m/z* 234.9932 (calcd for C_9_H_9_Cl_2_O_3_, [M + H]^+^, 234.9929), NMR: Table 1.

3-(3,5-Dichloro-4-hydroxyphenyl)propanoic acid methyl ester (**2**). Colorless solid; UV (MeOH) λ_max_ (log ε): 229 (4.15), 315 (4.17) nm; IR (film) ν_max_ 3335, 3030, 1665, 1305, 1219, 937 cm^−1^; HRESIMS *m/z* 249.0088 (calcd for C_10_H_11_Cl_2_O_3_, [M + H]^+^, 249.0085); NMR: Table 1.

3-(3-Chloro-4-hydroxyphenyl)propanoic acid (**3**). Colorless solid; UV (MeOH) λ_max_ (log ε): 225 (3.95), 307 (4.00) nm; IR (film) ν_max_ 3341, 3030, 1702, 1303, 1221, 936 cm^−1^; HRESIMS *m/z* 201.0321 (calcd for C_9_H_10_ClO_3_, [M + H]^+^, 201.0318), NMR: Table 1.

3-Phenylpropanoic acid (**4**). Colorless solid; ESIMS *m/z* 151.07 (C_9_H_11_O_2_, [M + H]^+^), NMR data: ^1^H NMR (850 MHz, CDCl_3_) δ_H_: 7.22 (2H, m, H-2,6), 7.30 (2H, m, H-3,5), 7.21 (1H, m, H-4), 2.97 (t, *J* = 7.5 H_2_-7), 2.69 (t, *J* = 7.5 H_2_-8); ^13^C NMR (312 MHz, CDCl_3_) δ_C_: 140.1 (qC, C-1), 128.5 (CH, C-2,6), 128.2 (CH, C-3,5), 126.4 (CH, C-4), 29.7 (CH_2_, C-7), 34.7 (CH_2_, C-8), 171.6 (qC, C-9).

*E*-Cinnamic acid (**5**). Colorless solid; ESIMS *m/z* 149.06 (C_9_H_9_O_2_, [M + H]^+^); NMR data: ^1^H NMR (850 MHz, CDCl_3_) δ_H_: 7.55 (2H, m, H-2,6), 7.41 (2H, m, H-3,5), 7.40 (1H, m, H-4), 7.79 (1H, d, *J* = 16.1 Hz, H-7), 6.45 (1H, d, *J* = 16.1 Hz, H-8); ^13^C NMR (312 MHz, CDCl_3_) δ_C_ 134.4 (C-1), 128.1 (C-2,6), 128.9 (C-3,5), 130.8 (C-4), 147.2 (C-7), 117.0 (C-8), 171.6 (C-9).

Cyclo(l-Phe-trans-4-OH-l-Pro) (**6**). Colorless solid; [α]_D_ −55° (c 0.10, MeOH); UV (MeOH) λ_max_ (log ε): 230 (4.10), 305 (4.07) nm; IR (film) ν_max_ 3451, 1663, 1629 cm^−1^; HRESIMS *m/z* 261.1241 (calcd for C_14_H_17_N_2_O_3_, [M + H]^+^, 261.1239), NMR: Table 2.

Cyclo(l-Phe-cis-4-OH-d-Pro) (**7**). Colorless solid; [α]_D_ +38° (c 0.10, MeOH); UV (MeOH) λ_max_ (log ε): 230 (4.10), 305 (4.07) nm; IR (film) ν_max_ 3450, 1662, 1629 cm^−1^; HRESIMS m/z 261.1241 (calcd for C_14_H_17_N_2_O_3_, [M + H]^+^, 261.1239), NMR: Table 2.

### 3.7. Antimicrobial Evaluation of Compounds ***1***–***7***

#### 3.7.1. Disc Diffusion Assay

Using the disc diffusion assay, the antimicrobial effects of **1**–**7** were evaluated at 100 μg/disc against several pathogenic microbes including *E. coli* (ATCC 25922), *C. albicans* (ATCC 14053), and *S. aureus* (ATCC 25923) and as previously described [48,49,50]. Ciprofloxacin (5.0 μg/disc) and ketoconazole (50 μg/disc) were used as positive antibiotics.

#### 3.7.2. Determination of the MIC of **1**–**7**

The MIC values of the compounds was evaluated using a macrodilution method [51]. Briefly, MeOH was used to dissolve the compounds at a final concentration of 2000 µg/mL, while distilled water was used to dissolve ciprofloxacin and ketoconazole at final concentrations of 100 µg/mL. All solutions were sterilized using syringe filters (0.2 µm). A two-fold serial dilution of the solutions was used in Mueller Hinton Broth (*MHB*) to afford concentrations between 1.0 and 1000 µg/mL for the compounds and between 0.125 and 64 µg/mL for ciprofloxacin and ketoconazole. From the 10^6^ colony-forming units (CFU)/mL microbial suspensions, 500 μL were added in sterile tubes giving inoculua of 5 × 10^5^ CFU/mL. Additional 100 μL of each stock solution of the compounds and antibiotics were added into the tubes. A control tube, which contains only the test microorganisms and methanol was prepared. The MeOH displayed no antimicrobial effect. Incubation of the tubes was accomplished at 37 °C for 48 h. The lowest concentrations of the compounds/antibiotics, which show no microbial growth were considered as MIC.

## 4. Conclusions

Purification of the antimicrobial fractions of the culture of the *Streptomyces coelicolor* LY001 afforded three chlorinated 3-phenylpropanoic acid derivatives including 3-(3,5-dichloro-4-hydroxyphenyl)propanoic acid (**1**), 3-(3,5-dichloro-4-hydroxyphenyl)propanoic acid methyl ester (**2**), and 3-(3-chloro-4-hydroxyphenyl)propanoic acid (**3**) along with 3-phenylpropanoic acid (**4**), E-cinnamic acid (**5**), and the dipeptides cyclo(l-Phe-*trans*-4-OH-l-Pro (**6**) and cyclo(l-Phe-*cis*-4-OH-d-Pro) (**7**). Structures of **1**–**7** were determined from interpretation of their NMR and HRESIMS spectroscopic data. The chlorinated 3-phenylpropanoic acid derivatives (**1**–**3**) showed selective and significant antimicrobial activities against *S. aureus* and *E. coli* and displayed modest effects towards *C. albicans*. On the other hand, the diketopiperazine alkaloids **6** and **7** were more active against and selective against *C. albicans* and less active than *E. coli* and *S. aureus.* Interestingly, this is the first report about natural occurrence of chlorinated 3-phenypropionic acid derivatives from a cultured marine-derived *Streptomyces*. These results suggest insight into the biosynthetic capacities of the cultured marine actinomycetes. Thus, compounds **1**–**3**, **6**, and **7** represent interesting compounds for the design of novel and effective antibiotic drugs.

## Figures and Tables

**Figure 1 marinedrugs-18-00450-f001:**
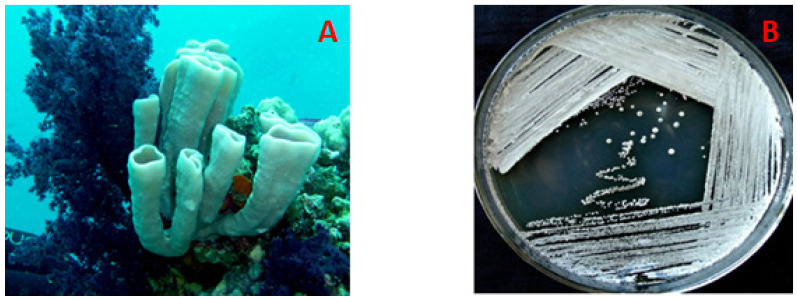
The Red Sea sponge *Callyspongia siphonella* (**A**) and the actinomycete *Streptomyces coelicolor* LY001 (**B**).

**Figure 2 marinedrugs-18-00450-f002:**
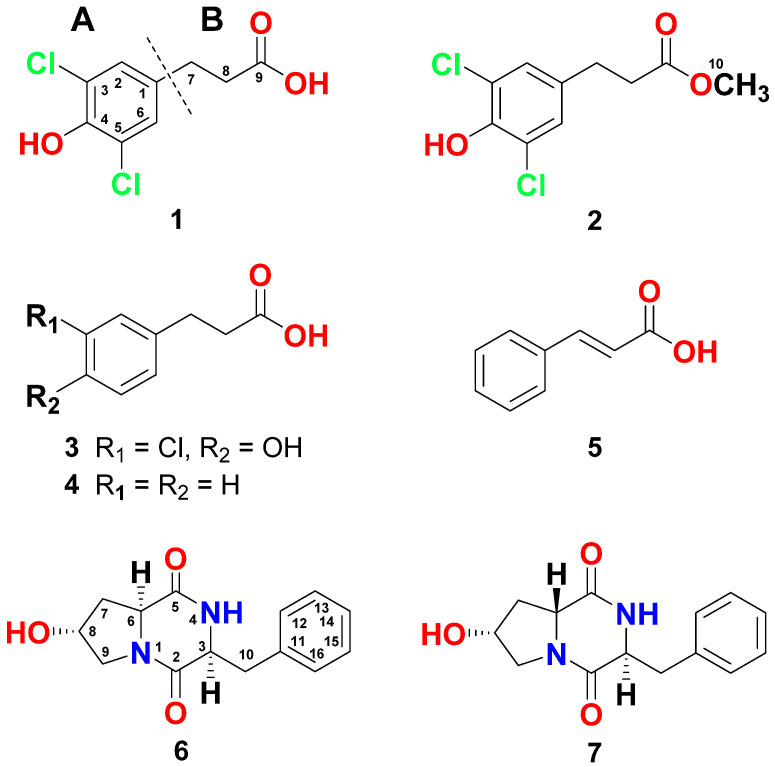
Structures of **1**–**7**.

**Figure 3 marinedrugs-18-00450-f003:**
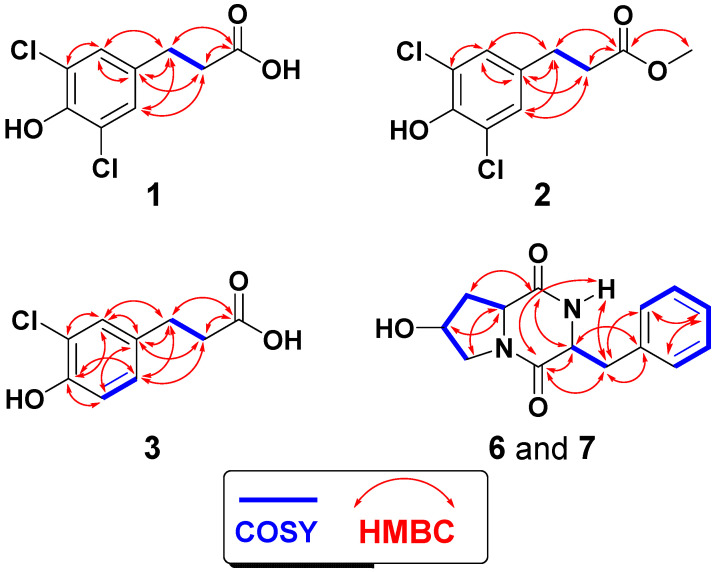
^1^H-^1^H COSY and ^1^H-^13^C HMBC of **1-3**, **6**, and **7**.

**Table 1 marinedrugs-18-00450-t001:** NMR data of **1–3** (CDCl_3_, ^1^H at 850 MHz, ^13^C at 213 Hz).

Position	1		2		3	
	δ_C_ (mult.) ^a^	δ_H_ (mult., *J* in Hz)	δ_C_ (mult.) ^a^	δ_H_ (mult., *J* in Hz)	δ_C_ (mult.) ^a^	δ_H_ (mult., *J* in Hz)
1	133.6, qC		133.8, qC		133.5, qC	
2	128.1, CH	7.12 (s)	128.1, CH	7.10 (s)	128.6, CH	7.17 (d, 2.5)
3	122.1, qC		121.0, qC		120.8, qC	
4	146.9, qC		146.6, qC		151.7, qC	
5	122.1, qC		121.0, qC		116.1, CH	6.93 (d, 8.5)
6	128.1, CH	7.12 (s)	128.1, CH	7.10 (s)	128.4, CH	7.02 (dd, 8.5, 2.5)
7	29.3, CH_2_	2.86 (t, 7.6)	29.8, CH_2_	2.84 (t, 7.6)	29.5, CH_2_	2.86 (t, 7.6)
8	34.5, CH_2_	2.64 (t, 7.6)	35.3, CH_2_	2.59 (t, 7.6)	34.8, CH_2_	2.65 (t, 7.6)
9	173.8, qC		173.3, qC		173.9, qC	
10			51.9, CH_3_	3.67 (s)		

^a^ Multiplicities of the ^13^C NMR signals were assigned from HSQC experiment.

**Table 2 marinedrugs-18-00450-t002:** NMR data of **6** and **7** (CDCl_3_).

Position	6 (^1^H at 600 MHz, ^13^C at 150 MHz)		7 (^1^H at 850 MHz, ^13^C at 213 Hz)	
	δ_C_ (mult.) ^a^	δ_H_ (mult., *J* in Hz)	δ_C_ (mult.) ^a^	δ_H_ (mult., *J* in Hz)
2	165.0, qC		165.2, qC	
3	56.1, CH	4.33 (dd, 10.8, 3.0)	59.1, CH	4.23 (tt, 4.2)
4 (N*H*)		5.60 (brs)		5.80 (brs)
5	169.5, qC		168.4, qC	
6	57.3, CH	4.46 (dd, 10.8, 6.0)	55.8, CH	3.17 (t, 8.5)
7a	37.8, CH_2_	2.37 (dd, 13.8, 6.0)	38.0, CH_2_	2.37 (ddd,13.7, 8.5, 5.1)
7b		2.06 (ddd, 13.8, 11.4, 4.2)		2.20 (m)
8	68.4, CH	4.60 (t, 4.2)	68.1, CH	4.40 (quin, 4.0)
9a	54.4, CH_2_	3.80 (dd, 13.2, 4.2)	53.4, CH_2_	3.82 (dd, 12.7, 2.5)
9b		3.58 (d, 13.2)		3.35 (dd, 12.7, 5.1)
10a	36.6, CH_2_	3.63 (dd, 14.5, 4.2)	40.8, CH_2_	3.13 (dd, 14.0, 6.8)
10b		2.77 (dd, 15.5, 10.8)		3.10 (dd, 14.0, 4.2)
11	135.7, qC		135.2, qC	
12	129.0, CH	7.22 (d, 7.5)	129.7, CH	7.21 (d, 7.3)
13	129.3, CH	7.36 (t, 7.5)	128.9, CH	7.31 (t, 7.3)
14	127.6, CH	7.29 (t, 7.5)	127.6, CH	7.30 (t, 7.3)
15	129.3, CH	7.36 (t, 7.5)	128.9, CH	7.31 (t, 7.3)
16	129.0, CH	7.22 (d, 7.5)	129.7, CH	7.21 (d, 7.3)

^a^ Multiplicities of the ^13^C NMR signals were assigned from HSQC experiment.

**Table 3 marinedrugs-18-00450-t003:** Antimicrobial effects of **1**–**7**.

Compound	*E. coli*		*S. aureus*		*C. albicans*	
	Inhibition Zone (mm)	MIC (µg/mL)	Inhibition Zone (mm)	MIC (µg/mL)	Inhibition Zone (mm)	MIC (µg/mL)
1	23	16	17	32	9	125
2	20	32	12	64	7	250
3	18	32	15	32	7	250
4	15	64	11	125	6	250
5	NT	NT	NT	NT	NT	NT
6	7	250	9	250	12	32
7	11	125	7	250	14	32
Ciprofloxacin ^a^	30	0.25	22	0.5	NT	NT
Ketoconazole ^b^	NT	NT	NT	NT	30	0.5

^a^ positive antibacterial drug; ^b^ positive antifungal drug.

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
