# Peer review of "Antimicrobial Chlorinated 3-Phenylpropanoic Acid Derivatives from the Red Sea Marine Actinomycete Streptomyces coelicolor LY001"

_marinedrugs, 2020, doi:10.3390/md18090450_

Round 1

Reviewer 1 Report

This article by Shaala et al describes the isolation/purification and analysis of seven marine natural products which they analyze in appropriate spectral detail.  Three of the compounds are novel chloro derivatives of 3-phenylpropionic acid.  The isolated compounds are subsequently tested against S. aureus E. coli and C. albicans

This paper is a step forward on identifying additional bacterially produced chlorinated phenylpropanoic acid derivatives from sea specimens. It also expands the scope of Red Sea Marine actinomycetes and their use as an antibacterial/antifungal agent.  This reviewer looks forward to seeing what other natural products are isolated from sponge symbionts and recommends acceptance after minor revisions to address the following points:

  1. In Figure 3, for compound 3, I understand what the Author’s are explaining by using the COSY highlight, but it is more appropriate to reference a long-range coupling being faintly observed in the COSY between the A and X protons that was confirmed by a small (2.5 Hz) coupling constant than to attempt to show a COSY coupling across a quaternary center with a blue bar. Please update the text and figure accordingly.
  2. In Figure 3, please use double-headed arrows for HMBC couplings and be extra precise with their drawing so that the figure is easy to interpret.     
  3. Line (41) Please add references for the statement on the few reported species of actinomycetes from the Red Sea.
  4. Line (47) synthesized is misspelled
  5. Line (55) Please omit “the” before “academia”
  6. The spectral analysis is very thoughtful and thorough, but please add structures to the SI for ease of reading.   
  7. Line (451) the journal title in reference 41 needs to be abbreviated.  
  8. Lines (320 - 349).  Unless the Journal requires it, the listing of the spectral data that are included in the supporting information could be abbreviated such as “the 1H NMR, 13 NMR, COSY, HSQC, HMBC, and HRMS data for compound 1-7 are available in the supporting information”

Author Response

Comment#1: In Figure 3, for compound 3, I understand what the Author’s are explaining by using the COSY highlight, but it is more appropriate to reference a long-range coupling being faintly observed in the COSY between the A and X protons that was confirmed by a small (2.5 Hz) coupling constant than to attempt to show a COSY coupling across a quaternary center with a blue bar.

Our reply: The figure was fixed as requested by the reviewer.

Comment# 2: In Figure 3, please use double-headed arrows for HMBC couplings and be extra precise with their drawing so that the figure is easy to interpret. 

Our reply:   Double-headed arrows were used to display the HMBC correlations.

Comment# 3: Line (41) Please add references for the statement on the few reported species of actinomycetes from the Red Sea.

Our reply: A reference was added after this statement.

Comment# 4: Line (47) synthesized is misspelled

Our reply: The word “synthesized” was corrected.

Comment# 5: Line (55) Please omit “the” before “academia”

Our reply: The word “the” was deleted

Comment# 6: The spectral analysis is very thoughtful and thorough, but please add structures to the SI for ease of reading.

Our reply: The structures of compounds 1-7 were added to the proton and carbon spectra of the compounds

Comment# 7: Line (451) the journal title in reference 41 needs to be abbreviated.  

Our reply: The journal Title was abbreviated.

Line# 8: Lines (320 - 349).  Unless the Journal requires it, the listing of the spectral data that are included in the supporting information could be abbreviated such as “the 1H NMR, 13 NMR, COSY, HSQC, HMBC, and HRMS data for compound 1-7 are available in the supporting information”

Our reply: The description of the supporting information was changed as requested by the reviewer.

Reviewer 2 Report

Three new compounds possess simple structure, and the antibacterial activity does not sound so intriguing. In my opinion, this paper lacks originality and novelty for publication of this journal. Also, there are some concerns about the structural elucidation (e.g. COSY data) and grammatical errors. It is regrettable that this paper is not accepted in the present state. I would recommend the authors to add some new findings (e.g. identification of biosynthetic genes related to new compounds and MIC data).

Author Response

Our reply:

We would like to thank the reviewer for his comments. Below is our reply to the comments raised by the reviewer:

  • The MIC data of the compounds were evaluated and added to Table 3 in the revised manuscript.
  • With regards to the identification of the biosynthetic genes related to the new compounds, we regret that due to the current health situation and the global crisis of Covid-19 we will not be able to do gene sequencing  or send samples abroad. In addition, this task was not a part of the objective of this study. However, a proposed biosynthetic pathway for the phenylpropanoids was proposed and an additional figure (Figure 4) was added to the manuscript with related references.

Reviewer 3 Report

The work of Lamiaa A. Shaala and collaborators “Antimicrobial Chlorinated 3-Phenylpropanoic Acid 2 Derivatives from the Red Sea Marine Actinomycete 3 Streptomyces coelicolor LY001” for Marine Drugs describes the obtention of several natural products of Red Sea sponges which have been poorly investigated, the elucidation of the structures using NMR spectroscopy and mass spectrometry and the evaluation of the antimicrobial activity in three different microorganisms. The aims of the work are clearly and efficiently pointed out and are interesting for synthetic chemists as well as biologists working in antibiotics. It shall be highlighted that some of the compounds are the first time to be reported in its natural occurrence and have been poorly studied as a thorough search in the databases and the authors indicate. For instance, only 14 reports of both compounds 1 and 2 are obtained using Scifinder database. In addition, the structural assignment is well explained and can be followed easily and clearly. Only to minor comments to increase the quality of the manuscript in the present form:

(1) At the figure 3, there are some bold bonds between C1 and C3/C6 which attempt to describe potential COSY correlations. These bold bonds must be removed since the C1 have not protons to be correlated with (C1, it’s a quaternary carbon).

(2) With regards to the supplementary material, some of the NMR spectra can be expanded for making easier to the reader the visualisation with the naked eye. For example, aromatic region of Figures S5, S11, S17 and S23).

(3) The authors can explain and add the steps of the obtention, classification and purification of the Red Sea sponges along the results since it’s important to the reader as well as include the detailed explanation in the Materials and Methods.

Author Response

We would like to thank the reviewer for his valuable comments. Below is our reply to the raised comments by the reviewer:

Comment# 1: At the figure 3, there are some bold bonds between C1 and C3/C6 which attempt to describe potential COSY correlations. These bold bonds must be removed since the C1 have not protons to be correlated with (C1, it’s a quaternary carbon).

Our reply: Figure 3 was corrected as requested by the reviewer.

Comment# 2: With regards to the supplementary material, some of the NMR spectra can be expanded for making easier to the reader the visualisation with the naked eye. For example, aromatic region of Figures S5, S11, S17 and S23).

Our reply: Expansions of the aromatic regions of the Figures S5, S11, S17, and S23 were in the updated supporting information.

Comment# 3: The authors can explain and add the steps of the obtention, classification and purification of the Red Sea sponges along the results since it’s important to the reader as well as include the detailed explanation in the Materials and Methods.

Our reply: Additional information were added. IN addition, corresponding references containing complete and detailed description of the sponge was added. However, the complete text can’t be added again here to avoid the plagiarism.

Round 2

Reviewer 2 Report

According to the comments, the authors improved the manuscript. I evaluate the point, but it became clearer that the antibacterial activity of new compounds is very weak and not so intriguing from the MIC data. Although the biosynthetic pathway was also proposed in the revised text, it is based on known findings. I feel that the authors need to demonstrate the biosynthetic genes of new compounds in the marine actinomycete strain as they mention this is the first example of the chlorinated phenylpropanoic acid family from a natural source. Regrettably, I would not be satisfied with this response. I would recommend the authors to submit this manuscript to other journals such as J Antibiot and Nat Product Commun.

Author Response

I would like to thank the reviewer for his report.

However, with respect to his point of view, I do not agree with his negative evaluation of the revised version of the manuscript for the second time and his insistence to reject the manuscript and the request to determine Biosynthetic genes for the chlorinated compounds.

In addition, Due to the current health circumstances and the pandemic COVID-19, we are not able to send biological samples now or soon to determine biosynthetic genes of the compounds.